## [Decision Letter · Decision Letter 0]

15 Feb 2024

PONE-D-23-32084Emphasis should be placed on identifying and reporting research priorities to increase research value: an empirical analysisPLOS ONE

Dear Dr. Fei,

Thank you for submitting your manuscript to PLOS ONE. After careful consideration, we feel that it has merit but does not fully meet PLOS ONE’s publication criteria as it currently stands. Therefore, we invite you to submit a revised version of the manuscript that addresses the points raised during the review process.

However, before proceeding with publication, we kindly request that you address the following issues:

English Review: Please conduct a thorough English review of your manuscript to ensure clarity, coherence, and grammatical accuracy. This includes checking for proper sentence structure, punctuation, and word choice. We recommend seeking assistance from a proficient English speaker or a professional language editing service if necessary.

Minor Changes: Additionally, please make the minor revisions suggested by the reviewers. These may include clarifications, corrections of typographical errors, or adjustments to improve the flow of the manuscript.

We look forward to receiving your revised manuscript.

Kind regards,

Jose A. Calvache, MD, MSc, PhD

Academic Editor

PLOS ONE

Journal Requirements:

Additional Editor Comments:

Dear authors,

I hope this letter finds you well. I am writing to you regarding the manuscript titled "Emphasis should be placed on identifying and reporting research priorities to increase research value: an empirical analysis" which you submitted to PLOS ONE. I want to thank you for choosing our journal as a platform for sharing your research findings.

After careful review by our editorial team and external reviewers, I am pleased to inform you that your manuscript has been accepted for publication pending minor revisions. We believe that your research contributes significantly to the field, and we appreciate the effort and dedication you have invested in this work.

However, before proceeding with publication, we kindly request that you address the following issues:

English Review: Please conduct a thorough English review of your manuscript to ensure clarity, coherence, and grammatical accuracy. This includes checking for proper sentence structure, punctuation, and word choice. We recommend seeking assistance from a proficient English speaker or a professional language editing service if necessary. PLOS suggests that you thoroughly copyedit our manuscript for language usage, spelling, and grammar. If you do not know anyone who can do this, you may wish to consider employing a professional scientific editing service. Whilst you may use any professional scientific editing service of your choice, PLOS has partnered with both American Journal Experts (AJE) and Editage to provide discounted services to PLOS.

Minor Changes: Additionally, please make the minor revisions suggested by the reviewers. These may include clarifications, corrections of typographical errors, or adjustments to improve the flow of the manuscript.

We believe that addressing these points will enhance the overall quality of your manuscript and facilitate the publication process. Once you have made the revisions, please submit the revised manuscript along with a detailed response to the reviewers' comments through our online submission system.

If you require any clarification or assistance during the revision process, please do not hesitate to contact us. We are here to support you and ensure a smooth publication experience.

Thank you once again for considering PLOS ONE for the dissemination of your research. We look forward to receiving your revised manuscript and working with you towards its publication.

Yours sincerely,

Reviewers' comments:

Reviewer's Responses to Questions

**Comments to the Author**

1. Is the manuscript technically sound, and do the data support the conclusions?

Reviewer #1: Yes

Reviewer #2: Yes

2. Has the statistical analysis been performed appropriately and rigorously? 

Reviewer #1: Yes

Reviewer #2: Yes

3. Have the authors made all data underlying the findings in their manuscript fully available?

Reviewer #1: Yes

Reviewer #2: Yes

4. Is the manuscript presented in an intelligible fashion and written in standard English?

Reviewer #1: No

Reviewer #2: Yes

5. Review Comments to the Author

Reviewer #1: This manuscript highlights deficiencies in the reporting of research recommendations within Chinese clinical guidelines through a direct comparison of randomly selected GRADE (n=45) and Chinese (n=90) guidelines. The work has been done methodologically with structure and checking by consensus of 2 independent data extractors and nicely reported in tables and figures. The work is primarily descriptive but a useful foundation upon which Chinese guideline committees can address areas for improvement and elevation of their work to international standards.

My main criticisms relate to the grammatical/English language structure that makes it hard to read/understand.

In particular

lines 69-74 - too long, 2 'buts - break with a full stop

lines 81-84

lines 267 - 'another studies'

lines 279-284 - too long

lines 285-284 - needs re-phrasing

lines 311-315 - too long

Reviewer #2: I find the manuscript novel and interesting.

They carry out a broad review of other studies on the topic and show the lack of information on this specific topic.

The report of research priorities is investigated in a thorough manner (search for systematic information).

The presentation of the results is precise and organized. Despite the basic statistical treatment, the graphic presentation of the data is very good.

6. PLOS authors have the option to publish the peer review history of their article (what does this mean?). If published, this will include your full peer review and any attached files.

Reviewer #1: **Yes: **Craig Anderson

Reviewer #2: **Yes: **Hugo A. Mantilla-Gutierrez

---

## [Author Response · Author response to Decision Letter 0]

22 Feb 2024

Dear Dr. Jose A. Calvache,  

On behalf of my co-authors, we thank you very much for giving us an opportunity to revise our manuscript. We appreciate editor and reviewers very much for their positive and constructive comments and suggestions on our manuscript entitled “Emphasis should be placed on identifying and reporting research priorities to increase research value: an empirical analysis”. (ID: PONE-D-23-32084).

We have studied reviewer’s comments carefully and have made revisions in the paper. We have tried our best to revise our manuscript according to the comments. Attached please find the revised version, which we would like to submit for your kind consideration.  

The editors and reviewers did not question the specific research content, so we did not modify the content. We did our best to polish the language. There are many language modifications, we cannot leave track to them all. Therefore, our file labeled 'Revised Manuscript with Track Changes' is consistent with the file labeled 'Manuscript'. We would be glad to respond to any further questions and comments that you may have.

We would like to express our great appreciation to you and reviewers for comments on our paper. Looking forward to hearing from you.  

Thank you and best regards.   

Yours sincerely,

Yutong Fei

Corresponding author:

Name: Yutong Fei

22 Feb., 2024

Centre for Evidence-Based Chinese Medicine, Beijing University of Chinese Medicine

Email: feiyt@bucm.edu.cn

---

## [Editor Report · Decision Letter 1]

6 Mar 2024

Emphasis should be placed on identifying and reporting research priorities to increase research value: an empirical analysis

PONE-D-23-32084R1

Dear Dr. Fei,

We’re pleased to inform you that your manuscript has been judged scientifically suitable for publication and will be formally accepted for publication once it meets all outstanding technical requirements.

Kind regards,

Jose A. Calvache, MD, MSc, PhD

Academic Editor

PLOS ONE

Additional Editor Comments (optional):

Acceptance of Your Submitted Paper

Dear Dr. Yutong Fei,

I am delighted to inform you that your paper titled "Emphasis should be placed on identifying and reporting research priorities to increase research value: an empirical analysis" has been accepted for publication in PLOS ONE. Congratulations on this significant achievement!

The reviewers and editorial team have thoroughly evaluated your submission and found it to be of high quality, making a valuable contribution. Your research methodology, analysis, and conclusions have been commended for their rigor and relevance.

We believe that your paper will make a meaningful impact on our readership and contribute to advancing knowledge in the field. Your dedication to producing quality research is truly commendable, and we are honored to have your work featured in our journal.

Thank you once again for choosing PLOS ONE as the platform for sharing your research findings. We look forward to your continued contributions and to seeing your paper online.

Warm regards,